# Treatment Strategy for Intermediate-Stage Hepatocellular Carcinoma: Transarterial Chemoembolization, Systemic Therapy, and Conversion Therapy

**DOI:** 10.3390/cancers15061798

**Published:** 2023-03-16

**Authors:** Takeshi Hatanaka, Yutaka Yata, Atsushi Naganuma, Satoru Kakizaki

**Affiliations:** 1Department of Gastroenterology, Gunma Saiseikai Maebashi Hospital, Maebashi 371-0821, Japan; 2Department of Gastroenterology, Hanwa Memorial Hospital, Osaka 558-0041, Japan; 3Department of Gastroenterology, National Hospital Organization Takasaki General Medical Center, Takasaki 370-0829, Japan; 4Department of Clinical Research, National Hospital Organization Takasaki General Medical Center, Takasaki 370-0829, Japan

**Keywords:** hepatocellular carcinoma, intermediate-stage hepatocellular carcinoma, molecular-targeted therapy, tyrosine kinase inhibitor, immune checkpoint inhibitor, atezolizumab plus bevacizumab, durvalumab plus tremelimumab, transarterial chemoembolization, conversion therapy, up-to-7 criteria, TACE refractory, TACE inappropriate

## Abstract

**Simple Summary:**

Systemic therapy for advanced-stage hepatocellular carcinoma (HCC) is undergoing major changes owing to the advancements made in molecular-targeted therapies and immune checkpoint inhibitors (ICIs). Although transarterial chemoembolization has been used as the standard treatment for intermediate-stage HCC, it has recently included molecular-targeted drugs and ICIs. As intermediate-stage HCC encompasses a wide variety of HCCs, the appropriate regimen to be used and the order of drug administration, including the use of anti-angiogenic inhibitors, remain controversial. This review discusses treatment strategies for intermediate-stage HCC considering the background of progress in the treatment of advanced-stage HCC.

**Abstract:**

Transarterial chemoembolization (TACE) has been standard treatment for intermediate-stage hepatocellular carcinoma (HCC). However, all intermediate-stage HCC patients did not benefit from TACE treatment because intermediate-stage HCC encompasses a wide variety of HCCs. Owing to remarkable progress in systemic therapy, including molecular-targeted therapy for advanced-stage HCC, the standard treatment of HCC has recently shifted to systemic therapy. However, it remains controversial as to which treatment should be initially performed for intermediate-stage HCC. In addition, although curative treatment can be considered when the tumor shrinks, the timing of conversion therapy remains uncertain. This review summarizes the advances of HCC treatment and discusses treatment strategies for intermediate-stage HCC.

## 1. Introduction

Hepatocellular carcinoma (HCC) is the most common type of primary liver cancer and the fourth leading cause of cancer-related deaths worldwide [1]. Systemic chemotherapy including molecular-targeted drugs is an established treatment for various advanced-stage tumors. Treatment with sorafenib [2,3], regorafenib [4], lenvatinib [5], ramucirumab [6], and cabozantinib [7] has significantly improved the survival of patients with HCC. However, despite the great advances in targeted drugs, HCC has an extremely poor prognosis owing to the occurrence of drug resistance, relapses, and metastases. Recently, the use of new therapeutic strategies such as cancer immunotherapy has extended the lives of cancer patients.

Currently, the combination of an immune checkpoint inhibitor (ICI) and vascular endothelial growth factor (VEGF) inhibitors, atezolizumab (Atez) and bevacizumab (Bev), has been used as a first-line treatment for advanced-stage HCC [8]. A combination of durvalumab (Dur; anti-programmed cell death ligand 1(PD-L1) antibody) and tremelimumab (Tre; anti-cytotoxic T-lymphocyte-associated antigen-4 (CTLA-4) antibody) could be used as an alternative option [9]; several clinical trials combining various ICIs or ICIs with other novel agents are underway. However, the choice of regimen as a first-line treatment for intermediate-stage HCC remains controversial.

To date, transarterial chemoembolization (TACE) has been used as standard treatment for intermediate-stage HCC; recently, it was replaced by molecular-targeted drugs.

Intermediate-stage HCC includes a wide variety of tumors and affects the hepatic functional reserves. Therefore, according to the subclassification of intermediate-stage HCC, pharmacotherapy was considered in cases where TACE was considered ineffective. Additionally, the concepts of TACE-refractory and TACE-inappropriate were proposed, and the hepatic functional reserve decreased with repeated TACE. Recently, drug therapy has been introduced at an early stage prior to the administration of TACE, which is less effective. Furthermore, remarkable evidence supports the efficacy of combination therapy with TACE plus molecular-targeted drugs and cancer immunotherapy. This review aimed to summarize the advances made in HCC treatment and to discuss promising treatment strategies for intermediate-stage HCC.

## 2. The Definition and Standard Treatment of Intermediate-Stage HCC

In the Barcelona clinic liver cancer (BCLC) staging system, HCC is classified by tumor conditions (tumor number, maximum tumor size, the presence of vascular invasion, and the presence of extrahepatic lesions), liver reserve (Child–Pugh classification), and the Eastern Cooperative Oncology Group performance status (ECOG PS), and is divided into five stages: very early, early, intermediate, advanced, and terminal [10,11]. Intermediate-stage HCC was defined as the presence of four or more lesions, with liver functional preservation (Child–Pugh grade A or B), the absence of cancer-related symptoms (PS 0), and the absence of vascular invasion or extrahepatic spread beyond the BCLC-A criteria.

Although TACE has been used as the standard treatment for intermediate-stage disease [10,11], its treatment strategy has been reconsidered owing to the advances made in pharmacotherapy for advanced-stage HCC.

### 2.1. Effectiveness of TACE in Intermediate-Stage HCC Treatment

In 2000, two randomized controlled trials (RCTs) comparing TACE and control groups were published. Lo et al. compared 40 randomized TACE patients (pts) with 39 control pts. This study reported 1-, 2-, and 3-year survival rates of 57%, 31%, and 26% in the TACE group and 32%, 11%, and 3% in the control group (relative risk 0.50, 95% confidence interval [CI]: 0.31–0.81, *p* = 0.005) [12]. Llovet et al. randomized 112 pts into three groups: 37 for transarterial embolization (TAE), 40 for TACE, and 35 for symptomatic therapy. The TACE group had a significantly better prognosis (*p* = 0.009) [13]. Cammà et al. also reported the results of a meta-analysis of RCTs comparing TACE with best supportive care (BSC) and found that the 2-year mortality rate was significantly lower in the TACE group compared with that in the BSC group (odds ratio: 0.54, 95% CI: 0.33–0.89, *p* = 0.015) [14].

Existing evidence supports the use of TACE for the treatment of intermediate-stage HCC, and the results of an RCT reported 20 years ago have been adopted in various guidelines. As this RCT excluded patients with poor hepatic reserve, such as those with Okuda stage III and Child–Pugh class C, patients with intermediate-stage HCC are considered eligible for TACE treatment, and various relevant guidelines have been standardized [10,15,16]. In the Japanese guidelines, patients with Child–Pugh class A or B and multiple tumors with a diameter of more than 3 cm, two to three tumors, or multiple tumors (four or more) of any size are eligible for TACE treatment. Lencioni et al. reported a comprehensive systematic review of the lipiodol TACE with a response rate of 52.5% and a median survival time of 19.4 months [17]. However, as it was a long-term study (33 years), and the results might be influenced by the dilution bias due to the target period and subject cases selected and bias due to the variations in TACE technique, the results should be interpreted with caution [18].

### 2.2. Subclassification of Intermediate-Stage HCC

Intermediate-stage HCC involves a wide variety of HCCs with different pathologies, various tumor factors, and a reduction in hepatic functional reserve. Thus, Bolondi et al. proposed that intermediate-stage HCC should be divided into four substages (B1–B4) based on the following aspects: whether it is within or outside the up-to-7 criteria, which assess whether the sum of the number of tumors and the main tumor diameter exceeded 7; the Child–Pugh score; and the ECOG PS score [19]. The B1 stage was defined as an ECOG PS score of 0 with a Child–Pugh score of 7 points or less and within the up-to-7 criteria; TACE is the first-line treatment, and liver transplantation and combination therapy TACE with ablation are recommended as the alternative treatments [19]. Since then, several subclassifications have been proposed to refine this substage [20,21,22]. As the prognostic factors were four tumors and a tumor diameter of 7 cm, Yamakado et al. proposed the 4-of-7-cm criteria for the B1 stage (Child–Pugh score of 5–6 points and within 4-of-7-cm criteria), the B2 stage (Child–Pugh score of 7–8 points and 4-of-7-cm within criteria or Child–Pugh score of 5–8 points and 4-of-7-cm outside criteria), and the B3 stage (Child–Pugh score 9 points). With respect to the treatment, TACE should be performed in patients in B2 stage, while liver resection and local therapy are recommended in patients in B1 stage in addition to TACE. Meanwhile, patients in B3 stage are less likely to benefit from TACE [20]. Kudo et al. proposed the following modified Bolondi sub-classification, which was verified using a cohort: B1 stage (curative treatment) as having a Child–Pugh score of 5–7 points and meeting the up-to-7 criteria, B2 stage as having a Child–Pugh score of 5–7 points and meeting the up-to-7 criteria (non-radical, palliative treatment), B3-a as having a Child–Pugh score of 8 or 9 points and meeting the up-to-7 criteria (radical treatment), and B3-b as meeting the up-to-7 criteria only (palliative treatment or no treatment) [21,22]. These criteria have been used in the development of combination therapy using TACE and drug therapy for intermediate-stage HCC [23].

In relation to the treatment strategy by subclassification, TACE is considered effective in patients with a low tumor burden, but it is considered ineffective in those with a high tumor burden; hence, systemic therapy may be used as an alternative treatment (Figure 1).

### 2.3. Switching from TACE to Drug Therapy as Treatment for Intermediate-Stage HCC

Sorafenib, a tyrosine kinase inhibitor (TKI), has emerged as a drug therapy for HCC, and repeated TACE may lead to gradual deterioration of the liver function, which may cause Child–Pugh class A patients to miss the opportunity to receive sorafenib. Therefore, concepts of TACE refractory and TACE inappropriate have been proposed [24].

Inappropriate TACE is a state in which the treated blood vessel is damaged and the catheter itself cannot be inserted, thus hindering the performance of TACE. It is also characterized by the deterioration of liver reserve function to Child–Pugh class C after repeated treatment and major portal vein invasion, making TACE technically impossible owing to the presence of a huge A-P shunt. TACE refractory is defined as the presence of intrahepatic lesions that are difficult to control even if TACE is performed twice, despite changing the drug and reconsidering other blood vessels for treatment. Additionally, the levels of tumor marker either do not decrease or they persistently increase, which is indicative of vascular invasion or distant metastasis [24]. To date, switching from TACE to drug therapy is recommended for TACE refractory patients; however, there is no existing evidence to support the importance of switching from TACE to drug therapy. A retrospective study reported that sorafenib administration after TACE improved the prognosis compared with repeated TACE [25]. Kudo et al. used propensity score matching to compare 30 patients in the lenvatinib group and 60 patients in the TACE group with Child–Pugh class A, who were beyond the up-to-7 criteria for intermediate-stage HCC. The overall survival (OS) were 37.9 months in the lenvatinib group and 21.3 months in the TACE group (hazard ratio [HR]: 0.48, 95% CI: 0.10–0.35, *p* < 0.001) [25]. Moreover, a retrospective study reported that the hepatic reserve decreased with each repetition of TACE in patients who were beyond the up-to-7 criteria, and they were more likely to transition to Child–Pugh class B within a short period [26,27]. Therefore, the appropriate timing of switching from TACE to other treatments, such as molecular-targeted therapy, should be determined (Figure 1).

## 3. Advances in Cancer Immunotherapy

In HCC, the immunotherapy IMbrave150 (combination therapy with Atez and Bev) has shown positive results [8]. Subsequently, studies investigating other cancer immunotherapies, including COSMIC-312 [28], Checkmate 459 [29], KEYNOTE-240 [30], and the HIMALAYA trial, have been conducted [9]. The HIMALAYA trial compared the efficacy of combination therapy of Dur plus Tre with that of sorafenib [9]. The combination of Dur and Tre showed positive results compared with that of sorafenib treatment in this phase 3 trial [9]. These two regimens (Atez/Bev and Dur/Tre) have shown promising results and have been recommended as first-line treatments in the latest BCLC staging system [31]. However, post-progression TKIs have not yet been developed.

The Imbrave150 trial demonstrated that treatment with a combination of monoclonal antibody targeting VEGF (Bev) and anti-PD-L1 inhibitor (Atez) improved the patients’ survival compared with sorafenib treatment alone in previously untreated patients [8]. The HR for death in patients treated with Atez/Bev was 0.58 (95% CI: 0.42–0.79). The OS and progression-free survival (PFS) were well stratified, with Atez/Bev improving both outcomes and quality of life compared with sorafenib.

The HIMALAYA trial [9] revealed a significantly prolonged OS in the combination treatment group compared with that in the sorafenib group. Patients administered with Dur/Tre, the first approved combination therapy comprising anti-PD-L1 and anti-CTLA-4 antibodies for advanced-stage HCC, exhibited an HR of 0.76 (95% CI: 0.61–0.96). The COSMIC-312 trial compared the efficacy and safety of cabozantinib plus atezolizumab (Cab/Atez) with that of sorafenib as first-line therapy in patients with advanced-stage HCC [30]. The COSMIC-312 trial did not report an improvement in OS among patients receiving Cab/Atez.

The CheckMate 459 trial compared the efficacy of nivolumab monotherapy with that of sorafenib monotherapy in treatment-naïve patients. Nivolumab improved the OS, but the results were not significant [29]. Similarly, the KEYNOTE-240 trial compared the efficacy and safety of pembrolizumab monotherapy with that of the BSC in patients previously treated with sorafenib [30]. Pembrolizumab improved the OS and PFS. However, the results were not significant based on the specified criteria.

Selecting between Atez/Bev and Dur/Tre as a combination treatment remains a challenge. The criteria for using these two regimens as first-line therapies have not yet been established. One of the key points that need to be considered when making treatment decisions is whether anti-VEGF therapy is acceptable and tolerable.

VEGF induces several adverse events (AEs), including hypertension, proteinuria, thromboembolism, impaired wound healing, and bleeding in the gastrointestinal sites. The Imbrave150 trial reported that the most common Bev-related AEs were hypertension (31.0%), bleeding (25.2%), and proteinuria (21.3%). Screening for esophageal varices was performed prior to enrollment, and varices were treated as needed as bleeding from the gastrointestinal tract is a well-known AE associated with Bev treatment, sometimes leading to fatal events.

Proteinuria is also an important AE as post-progression TKI regimens such as sorafenib and lenvatinib inhibit the expression of VEGF; patients who develop Bev-associated proteinuria may require dose reduction or treatment discontinuation [32]. By contrast, Dur/Tre does not contain an anti-VEGF agent; hence, it may be preferred in patients who cannot tolerate anti-VEGF therapy. Furthermore, Atez/Bev may be preferred in patients with a high tumor burden. In the IMbrave150 trial, the PFS time was 6.8 months, while the response rate was 27.3%. In the HIMALAYA trial, patients who received the Dur/Tre treatment had a PFS time of 3.8 months and a response rate of 20.1%. Patients with primary portal vein tumor thrombosis (Vp4) were included in the IMbrave150 trial but were excluded from the HIMALAYA trial. The incidence of disease progression was lower in the Atez/Bev group compared with that in the Dur/Tre group (19.6% vs. 39.9%). Therefore, Atez/Bev may be preferred for patients with a high tumor burden, such as those with 50% liver involvement and portal vein tumor thrombosis.

Furthermore, Atez/Bev treatment may be preferred in patients with WNT/β-catenin mutations and non-viral infections. Anti-VEGF therapies, including Atez/Bev and TKI regimens, improve the tumor microenvironment and produce moderate responses. By contrast, Dur/Tre may be less effective in patients with WNT/β-catenin mutations and non-viral-associated HCC. Although increasing evidence has shown that systemic therapy is effective and safe for patients with advanced-stage HCC, the specific regimens that are suitable for first-line treatment remain unknown.

## 4. Combination of TACE and Molecular-Targeted Drugs

Systemic therapy using a highly responsive agent such as lenvatinib (LEN) in combination with subsequent selective TACE for residual tumors enhances the curative effect of TACE, preserves liver function, suppresses hypoxia-induced cytokines, and ultimately improves the patients’ survival. Therefore, the Asian Pacific Experts on Primary Liver Cancer (APPLE) [33] and the Japanese Society of Hepatology (JSH) [34] consensus statements recommend LEN as the first-line treatment for patients with intermediate-stage HCC who are not suitable for TACE. Recently, clinical practice guidelines from the E-updated European Society for Medical Oncology recommended upfront systemic therapy for patients who are ineligible for TACE [35].

Furthermore, the 2020 update of the American Society for Liver Consensus Statement recommends upfront systemic therapy in addition to TACE for patients with intermediate-stage HCC with a high tumor burden [36]. The revision of systemic therapy as a treatment option for intermediate-stage HCC was the first major revision carried out in 20 years since the establishment of the BCLC algorithm in 1999.

Thus, multiple studies have investigated the efficacy of TACE and molecular-targeted drug combination therapy in intermediate-stage HCC after the evidence on sorafenib efficacy was reported. Combination therapy with TACE and molecular-targeted drugs has two objectives: to delay tumor progression after TACE and to administer TACE in areas where molecular-targeted drugs are ineffective.

The combined effect of TACE and molecular-targeted drugs can be explained as follows: the molecular-targeted drug acts on residual tumors caused by TACE, exerting anti-angiogenic and anti-tumor growth effects to prevent the exacerbation of residual tumors, avoid the development of new intrahepatic lesions, and suppress vascular invasion and distant metastasis. Molecular-targeted drugs with an anti-VEGF effect act on tumor blood vessels to improve the vascular permeability; reduce the tumor interstitial pressure; improve the drug delivery; enhance the treatment efficacy [34]; and suppress the effect of the transient increase of VEGF immediately after the occurrence of TACE-related ischemia, which is considered as one of the causes of tumor exacerbation [33,34,37,38]. Although multiple clinical trials investigating the efficacy of combination therapy with TACE and molecular-targeted drugs have been conducted, their efficacy has not been demonstrated. Under these circumstances, the TACTICS trial (TACE therapy in combination with sorafenib), which investigated the combined effect of sorafenib and lipiodol TACE, showed the potential effects of treatment with a combination of targeted drugs and TACE.

The TACTICS trial was a multicenter, randomized, controlled trial that investigated PFS and OS in HCC patients who had previously received consecutive therapy, including sorafenib followed by on-demand selective TACE. In this study, the TACE-related PFS times (the primary endpoint) were 13.5 months for patients receiving TACE alone and 25.2 months for patients receiving sequential sorafenib plus TACE (HR: 0.59, 95% CI: 0.41–0.87, *p* = 0.006) [39,40,41]. In patients who developed tumors beyond the up-to-7 criteria, the PFS was 13.1 months longer in those treated with sorafenib-TACE sequential therapy compared with that in patients treated with TACE alone (22.1 months vs. 9.0 months, HR: 0.674). The OS was also prolonged by 11.3 months with the addition of sorafenib to TACE (36.3 months vs. 25.0 months, HR: 0.898) [40]. In patients within the up-to-7 criteria, sorafenib-TACE sequential therapy extended the PFS by 9.7 months compared with TACE alone (24.9 vs. 15.2 months, HR: 0.756) [40].

Sorafenib-TACE sequential therapy also extended the OS by 3.7 months compared with TACE alone (35.6 months vs. 31.9 months, HR: 0.924). These data indicate that sequential therapy with sorafenib and TACE was more effective than TACE alone in prolonging the PFS [40]. Although the TACTICS treatment improved the PFS, it did not significantly extend the OS. For this reason, 76.3% of the patients who received TACE alone also received post-trial treatment, and nearly half of these patients received sorafenib. Therefore, patients generally received aggressive post-trial treatment, which can extend the post-progression survival (PPS) and weaken the benefit obtained from the original trial treatment [41]. However, the median survival benefit of 5.4 months observed in the TACTICS trial was considered as a clinically reasonable result. TACTICS demonstrated the effectiveness of the initial treatment with an anti-VEGF inhibitor, followed by selective TACE, for intermediate-stage HCC; it is thought to be a landmark trial in the field of TACE combination trials.

Sequential LEN-TACE therapy yielded positive results. With respect to LEN-TACE sequential therapy, the OS was 37.9 months in patients treated with LEN followed by selective TACE, while it was 21.3 months in those treated with TACE alone; therefore, sequential LEN-TACE therapy significantly improved the patients’ OS (HR: 0.48; 95% CI: 0.16–0.79, *p* < 0.01) [40,42]. The PFS, the objective response rate (ORR) based on the modified response evaluation criteria in solid tumors (mRECIST), and liver function preservation improved in the LEN-treated group [42]. Furthermore, 5 of 30 (17%) patients who received sequential LEN-TACE therapy achieved a cancer-free status. The results have been replicated in other clinical studies [25], and LEN-TACE sequential therapy has become one of the established approaches for intermediate-stage HCC in Japan. Moreover, despite the advanced stage of HCC (macroscopic portal invasion, 71.8%; extrahepatic spread, 55.3%) reported in the LAUNCH trial, LEN-TACE was significantly better in terms of OS, PFS, ORR, and disease control rate (DCR) compared with LEN alone [43]. Consequently, LEN-TACE sequential therapy may be a potential option that can lead to a cancer-free status or a longer survival, even in patients with TACE-unsuitable intermediate-stage HCC who do not achieve complete response (CR).

## 5. New Treatment Strategy for Intermediate Stage HCC: TACE plus Cancer Immunotherapy, and Conversion Therapy

Currently, the development of new drugs for HCC focuses on establishing an effective cancer immunotherapy. With the advent of effective cancer immunotherapy, nivolumab monotherapy, pembrolizumab monotherapy, Atez/Bev combination therapy, and Dur/Tre combination therapy have been used as postoperative adjuvant chemotherapy for HCC, and their recurrence-suppressive effects after hepatic resection or ablation are being verified [44]. Thus, the possibility of prolonged PFS through the concomitant use of cancer immunotherapy for intermediate-stage HCC has been investigated.

Atez/Bev combination therapy yielded a high response rate in patients with intermediate-stage HCC (44% ORR per RECIST v1.1) [8,45,46]. Unlike targeted agents, the use of Atez/Bev combination therapy resulted in a significant tumor shrinkage, even in patients with highly malignant positron emission tomography-positive HCC [47], including those with confluent multinodular and poorly differentiated HCC. Consequently, surgical excision, ablation, or curative TACE is feasible, resulting in the achievement of pathological CR and drug-free status in 20–30% of patients [47]. In oncology, once systemic therapy is initiated, it is continued if the patient achieves a stable disease (SD). This is particularly true if a partial response (PR) is achieved. In such cases, treatment is unchanged as long as PR persists, especially in patients with intermediate-stage HCC that remains locally advanced without vascular invasion or extrahepatic spread, and ablation or curative treatment including resection is recommended. Other curative treatments, such as targeted TACE, can also be provided, and they may result in a significant tumor shrinkage. The effects of TACE may be improved by the VEGF-inhibitory action of Bev, or the effects of Atez may be enhanced by the release of cancer antigens induced by TACE. If significant tumor shrinkage is achieved using Atez/Bev combination therapy, definitive conversion therapy should be performed at the best possible moment rather than continuing the sequential systemic therapy.

If systemic therapy is continued, the appropriate timing and method of TACE enforcement should be determined as no clear criteria have been established related to these aspects, that is, whether TACE should be performed “before PD occurs”, “what is the best response”, whether TACE should be administered in “nodules that showed PD”, whether TACE should be used “as conventional-TACE or DEB-TACE”, etc. This is because meta-analytic assessment is hampered by the excessive heterogeneity of intermediated-stage HCC between trials [48]. In addition, the progression pattern of patients treated with sorafenib influences the prognosis, and new vascular invasions or new extrahepatic lesions correlate with the worst prognosis [49]. This finding indicates that the prognosis differs depending on the type of PD caused by systemic therapy [31]. Therefore, the timing of TACE during continuous systemic therapy should be determined based on treatment responsiveness and type of PD assessed through radiological imaging. In intermediate-stage HCC, which has a variety of treatment patterns (whether TACE should be administered in combination with a single drug or performed three times, and whether the drugs should be changed to four or more), the appropriate treatment should be selected after considering the changes in liver function and tumor marker values. When systemic therapy is used in combination with TACE, systemic therapy with the same drug is continued, while TACE is only used for an existing tumor that shows slowed growth or for a new solitary nodule that appears as an intra-hepatic lesion even in patients with PD; when multiple tumor recurrences, distant metastasis, or vascular invasion occur, switching to new systemic therapeutic agents might be required (Figure 2) [47,50].

Thus, the concept of systemic therapy for intermediate-stage HCC is completely different from that of conventional sequential therapy for advanced-stage HCC. Intermediate-stage HCC is treated by switching from TACE to drug therapy or by combining TACE and drug therapy (Figure 3).

HCC, hepatocellular carcinoma; TACE, transarterial chemoembolization; TKI, tyrosine kinase inhibitor; ICI, immune checkpoint inhibitor; mALBI, modified albumin-bilirubin grade; Atz, atezolizumab; Bev, bevacizumab; Dur, durvalumab; Tre, tremelimumab. PS: performance status, and RFA: radiofrequency ablation.

Several studies have been conducted to verify the combined effects of TACE and molecular-targeted drugs (Table 1) [51].

Pathologic CR is rarely achieved with systemic therapy alone, such as lenvatinib or Atez/Bev. Patients who appear to have achieved CR according to the mRECIST may have residual cancer after resection. In such cases, recurrence is likely when systemic therapy is discontinued. To prevent this, even if the imaging findings indicate CR or a good response, conversion therapy should be performed as radically as possible.

## 6. Conclusions

The concept of systemic therapy for intermediate-stage HCC differs from that for advanced HCC. As intermediate-stage HCC comprises various cancer conditions, it requires a unique approach that includes a combination of locoregional and systemic treatments. The goal of HCC treatment is to achieve a complete cure. Further studies with better evidence regarding the timing of treatment switching and concomitant therapy are warranted.

## Figures and Tables

**Figure 1 cancers-15-01798-f001:**
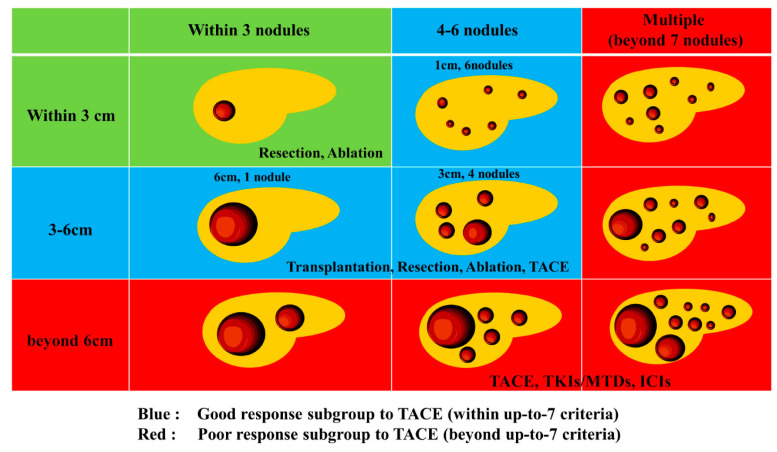
Heterogeneity and treatment strategy for intermediate stage HCC.

**Figure 2 cancers-15-01798-f002:**
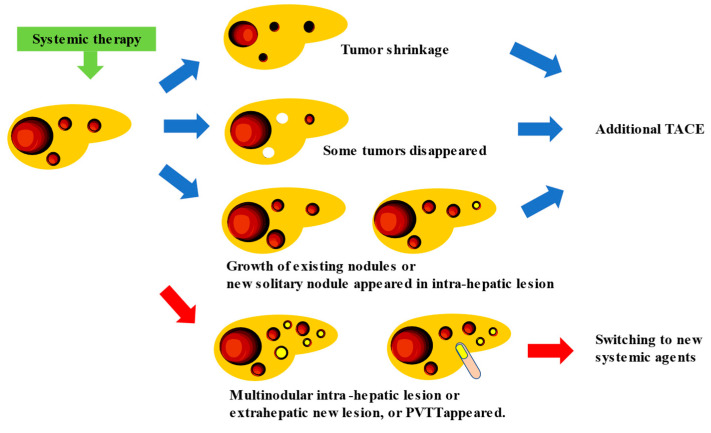
Proposal for the timing of additional TACE treatment for intermediate-stage HCC patients receiving systemic therapy. White tumors indicated disappeared lesions, and yellow ones showed PD lesions.

**Figure 3 cancers-15-01798-f003:**
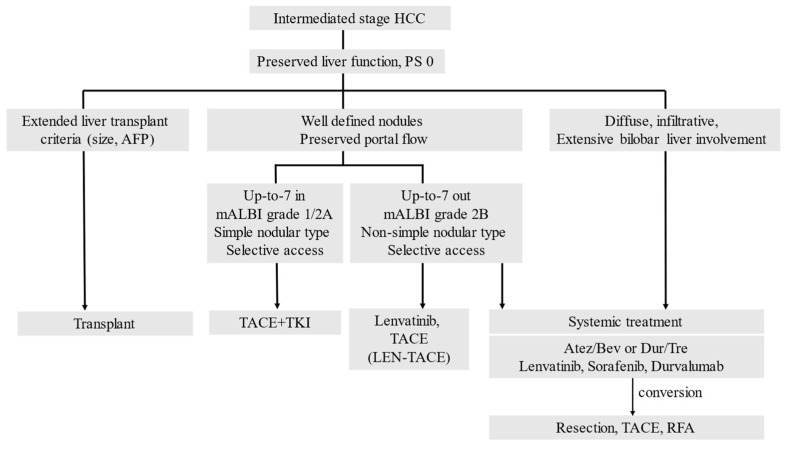
Treatment strategy for intermediate-stage HCC.

**Table 1 cancers-15-01798-t001:** Ongoing trials on combination therapy with transarterial chemoembolization (TACE) and immune-checkpoint inhibitors.

Trials	Subjects (Patients)	Treatments	Phase	Numbers	Evaluation Criteria	Primary End Points
EMERALD-1	CP score 5-7	Arm1: TACE+Dur+Placebo	III	710	RECIST v1.1	PFS
	ECOG PS 0-1	Arm2: TACE+Dur+Bev				
	No extra hepatic lesions	Arm3: TACE+Placebo				
	Unresectable HCC					
TACE-3	CP-A	Arm1: TACE and/or TAE	II	522	RECIST v1.1	TTTP
	ECOG PS 0-1	Arm2: TACE and/or TAE+Nivolumab	III			OS
	No extra hepatic					
	Unresectable HCC					
LEAP-012	CP-A	Arm1: TACE+Len+Pem	III	950	RECIST v1.1	PFS
	First treatment (naïve)	Arm2: TACE+Placebo				OS
	No extra hepatic					
	unresectable					
CheckMate 74W	Intermediate stage	Arm1: TACE+Nivolumab+Ipilimumab	III	765		TTTP
	ECOG PS 0-1	Arm2: TACE+Nivolumab+Placebo				OS
	BMU7	Arm3: TACE+Placebo				
NCT01909866	BCLC B/C	Arm1: TACE+Len+Camrelizumab	II	II 40		PFS
	ECOG PS 0-1		III	III		
	Untreated TKIs, ICIs					
TALENTACE	CP-A	Arm1: TACE+Atezo+Bev	III	342		TACE PFS
(NCT 04712643)	ECOG PS 0-1	Arm2: TACE				OS
	Untreated TKIs, ICIs					

TACE: transarterial chemoembolization, TAE: transarterial embolization, CP: Child-Pugh, ECOG: Eastern Cooperative Oncology Group, PS: performance status, PFS: progression free survival, TTTP: time to TACE progression, OS: overall survival, BMU7: beyond Milan and up-to-seven, BCLC: Barcelona Clinic Liver Cancer, PFS: progression free survival, TTTP: time to TACE progression, OS: overall survival, BMU7: beyond Milan and up-to-seven, BCLC: Barcelona Clinic Liver Cancer.

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
