# Peer review of "Treatment Strategy for Intermediate-Stage Hepatocellular Carcinoma: Transarterial Chemoembolization, Systemic Therapy, and Conversion Therapy"

_cancers, 2023, doi:10.3390/cancers15061798_

Round 1
Reviewer 1 Report
The paper by Hatanaka et al. is well written and I would like to congratulate with the authors for thoroughly reviewing recent treatment strategy for intermediate-stage hepatocellular carcinoma, a clinical stage with high heterogeneity. I think that it could be suitable for publication as it could provide some useful information about future treatment.
Author Response
Thank you so much for your precious comments.
Reviewer 2 Report
This is a review article focusing on treatment strategy for intermediate stage hepatocellular carcinoma(HCC). I have several comments.
1. The authors focused on the subclassification of intermediate stage HCC, however the meaning of this subclassification in treatment decision is not well documented.
2. The conclusion is difficult to understand. What treatment the authors recommend?
Author Response
1. The authors focused on the subclassification of intermediate stage HCC, however the meaning of this subclassification in treatment decision is not well documented.
→ Thank you so much for your precious comments. As you pointed out, due to the lack of description of treatment strategies by subclassification, we added the following sentence (with yellow highlight) in the end of 2.2 Subclassification of intermediate-stage HCC (P4, line1-3).
''Concluding of the treatment strategy by subclassification, a low tumor burden could be expected to have a therapeutic effect of TACE, but a high tumor burden could not be expected to treat with TACE, therefore, systemic therapy may be expected (Fig.1).''
2. The conclusion is difficult to understand. What treatment the authors recommend?
→Thank you so much for your precious comments. As you pointed out, the content is too long and the main points are difficult to understand, so the following part has been deleted and briefly described.
''Early switching from TACE, which has been the standard treatment, to another effective drug therapy is thought to improve the prognosis of intermediate-stage HCC due to advances in multiple drug treatments, including cancer immunotherapy. The TACTICS trial demonstrated that a sequential approach consisting of agents with anti-VEGF activity followed by selective TACE improved PFS and OS in Up-to-7 criteria-exceeding cases. This was confirmed by the improvement in OS with the LEN-TACE sequential therapy, which included first-line LEN, followed by selective TACE. Continuous LEN-TACE therapy has already been established and widely used in Japan [48]. Unlike other targeted therapies, Atez/Bev combination therapy induces tumor regression, even in patients with high-grade intermediate stage HCC, including poorly differentiated HCC.''
''Long-term drug-free patients are expected to survive without AEs and have a good quality of life. If HCC recurs, there are still many treatment options available. If it can be detected early with preserved liver function, definitive treatment can be applied again.''
In addition, we moved the sentence '’Intermediate-stage HCC is treated by switching from TACE to drug therapy or by combining TACE and drug therapy (Fig. 3).’' into the paragraph 2.4.3.(New treatment strategy for intermediate stage HCC: TACE plus cancer immunotherapy, and conversion therapy) and described the recommended treatment at the moment in Fig. 3.
Reviewer 3 Report
The Review article entitled" Treatment strategy for intermediate-stage hepatocellular carcinoma: transarterial chemoembolization, systemic therapy, and conversion therapy" is discussing the treatment strategies for intermediate-stage HCC against the background of progress in the treatment of advanced HCC. As a reviewer I have some serious observations which need to be addressed:
1. The abstract of the Review is very vague and does not convey the importance of the topic
2. There are lot of grammatical mistakes and spelling mistakes.
3. Overall most of the paragraphs are confusing they need to be rewritten properly.
4. The diagrams lack the clarity
5. Conclusion needs to be written properly
Author Response
Dear reviewer
We would like to thank the reviewers for the insightful comments to our manuscript ‘cancers-2245641’. We have modified our manuscript to reflect the suggestions provided by the reviewers.
Here is a point-by-point response to the reviewer3s’ comments and concerns.
- The abstract of the Review is very vague and does not convey the importance of the topic
Response
Thank you for your insightful comment. Following your comment, we have changed the entire abstract to focus on the importance of the topic.
- There are lot of grammatical mistakes and spelling mistakes.
Response
Thank you for your insightful comment. Following your comments, we double-checked the entire text and asked a native English speaker to proofread the English. We also attached their certification.
- Overall most of the paragraphs are confusing they need to be rewritten properly.
Response
We thank you for your insightful comment.
Following your comments, we changed some of the paragraphs to be more understandable by readers.
- The diagrams lack the clarity
Response
We thank you for your insightful comment.
Following your comments, we modified Fig.2 and Fig.3 to be understood clearly.
- Conclusion needs to be written properly
Response
We thank you for your insightful comment. Following your comments, we changed conclusion to make it simpler.

Reviewer 4 Report
The author provided a comprehensive view about the treatment strategies for inter-mediate-stage advanced hepatocellular carcinoma (HCC). This manuscript was well written. I have no other constructive suggestions for the further improvement. I hope the author could shorten the conclusion part, which was now too long and reference number (like [48], fig3) should be also avoided to get a short summary and prospect.
Author Response
The author provided a comprehensive view about the treatment strategies for inter-mediate-stage advanced hepatocellular carcinoma (HCC). This manuscript was well written. I have no other constructive suggestions for the further improvement. I hope the author could shorten the conclusion part, which was now too long and reference number (like [48], fig3) should be also avoided to get a short summary and prospect.
→Thank you so much for your precious comments. As you pointed out, the content is too long and the main points are difficult to understand, so the following part has been deleted and briefly described.
''Early switching from TACE, which has been the standard treatment, to another effective drug therapy is thought to improve the prognosis of intermediate-stage HCC due to advances in multiple drug treatments, including cancer immunotherapy. The TACTICS trial demonstrated that a sequential approach consisting of agents with anti-VEGF activity followed by selective TACE improved PFS and OS in Up-to-7 criteria-exceeding cases. This was confirmed by the improvement in OS with the LEN-TACE sequential therapy, which included first-line LEN, followed by selective TACE. Continuous LEN-TACE therapy has already been established and widely used in Japan [48]. Unlike other targeted therapies, Atez/Bev combination therapy induces tumor regression, even in patients with high-grade intermediate stage HCC, including poorly differentiated HCC.''
''Long-term drug-free patients are expected to survive without AEs and have a good quality of life. If HCC recurs, there are still many treatment options available. If it can be detected early with preserved liver function, definitive treatment can be applied again.''
In addition, we moved the sentence '’Intermediate-stage HCC is treated by switching from TACE to drug therapy or by combining TACE and drug therapy (Fig. 3).’' into the paragraph 2.4.3. (New treatment strategy for intermediate stage HCC: TACE plus cancer immunotherapy and conversion therapy).
Round 2
Reviewer 2 Report
The authors have revised the manuscript appropriately.
Author Response
We thank you for your insightful comment.
Reviewer 3 Report
The manuscript entitled" Treatment strategy for intermediate-stage hepatocellular carcinoma: transarterial chemoembolization , systemic therapy, and conversion therapy" has been revised properly as per the given suggestion.